# Surface-kinetics mediated mesoporous multipods for enhanced bacterial adhesion and inhibition

Tiancong Zhao[1], Liang Chen[1], Peiyuan Wang[1], Benhao Li[1], Runfeng Lin[1], Areej Abdulkareem Al-Khalaf[2], Wael N. Hozzein [3,4], Fan Zhang [1], Xiaomin Li [1]* & Dongyuan Zhao [1]*

Despite the importance of nanoparticle's multipods topology in multivalent-interactions enhanced nano-bio interactions, the precise manipulation of multipods surface topological structures is still a great challenge. Herein, the surface-kinetics mediated multi-site nucleation strategy is demonstrated for the fabrication of mesoporous multipods with precisely tunable surface topological structures. Tribulus-like tetra-pods $Fe_3O_4$@$SiO_2$@RF&PMOs (RF = resorcinol-formaldehyde resin, PMO = periodic mesoporous organosilica) nanocomposites have successfully been fabricated with a centering core@shell $Fe_3O_4$@$SiO_2$@RF nanoparticle, and four surrounding PMO nanocubes as pods. By manipulating the number of nucleation sites through mediating surface kinetics, a series of multipods mesoporous nanocomposites with precisely controllable surface topological structures are formed, including Janus with only one pod, nearly plane distributed dual-pods and tri-pods, three-dimensional tetrahedral structured tetra-pods, etc. The multipods topology endows the mesoporous nanocomposites enhanced bacteria adhesion ability. Particularly, the tribulus-like tetra-pods mesoporous nanoparticles show ~100% bacteria segregation and long-term inhibition over 90% after antibiotic loading.

[1] Department of Chemistry and Laboratory of Advanced Materials, State Key Laboratory of Molecular Engineering of Polymers, Collaborative Innovation Center of Chemistry for Energy Materials (2011-iChEM), Fudan University, Shanghai 200433, PR China. [2] Biology Department, College of Science, Princess Nourah Bint Abdulrahman University, Riyadh, Saudi Arabia. [3] Bioproducts Research Chair, Zoology Department, College of Science, King Saud University, Riyadh 11451, Saudi Arabia. [4] Botany and Microbiology Department, Faculty of Science, Beni-Suef University, Beni-Suef, Egypt. *email: lixm@fudan.edu.cn; dyzhao@fudan.edu.cn

There are many characters worth learning from intriguing natural creations for developing functional materials with versatile applications[1–5], especially the surface topological structure enhanced interfacial interactions in natural world. For example, tribulus seeds usually have a few sharp rigid burs on the surface, i.e., multipods shape, ensuring that the seeds can be easily attached onto the fur of animals for spreading[6]. Viruses can invade cell membranes through their multiple pod-like spikes[7–9]. Similarly, it has been widely recognized that the strong interaction between nanoparticles and bio-hosts is crucial for intracellular drug delivery, bacterial inhibition, and so on[10–12], which is quite related to the surface topology induced multivalent nano-bio interactions[13–15]. Like the tribulus seeds and viruses, the multipods surface structure can provide nanomaterials with multivalent interactions toward biological interfaces, thus achieving higher binding strength, enhanced adherence and penetration into biological hosts[16–18]. However, it is still a great challenge to fabricate such multipods structured nanoparticles with precisely controlled surface topological structures. Furthermore, most of the reported multipods nanomaterials are based on dense crystals[19–23], polymers[24,25], etc., which cannot provide enough storage space for functional guest molecule loading, restricting the further practical applications of the nanomaterials with multipods surface topological structures.

To obtain multipods nanoparticles with both precisely controllable surface topological structure and large storage space for functional molecules loading, the controllable multi-site nucleation of porous pods during nanoparticles' surface manufacturing plays a crucial role[26,27]. Recently, the anisotropic island nucleation and growth strategy have been developed for the fabrication of various intriguing asymmetric mesoporous nanomaterials, such as Janus nanoparticles[28–30], eccentric single-hole nanorattles[31], one-dimensional (1D) diblock and triblock nanocomposites[32], etc. However, the number of nucleation sites for the mesoporous pods assembly has been restricted to one for the fabrication of asymmetric dimer structure, and the island nucleation and growth of these pods are randomly distributed. It is still a great challenge to precisely control the number and distribution of nucleation sites for the fabrication of multipods mesoporous nanoparticles with controllable surface topological structures.

Herein, the surface-kinetic mediated multi-site nucleation strategy is demonstrated to precisely control the nucleation number of periodic mesoporous organosilicas (PMOs) on the surface of resorcinol-formaldehyde resin (RF). The tetra-pods $Fe_3O_4@SiO_2@RF\&PMOs$ (RF = resorcinol-formaldehyde resin) mesoporous nanocomposites have been fabricated (Fig. 1). The multipods are made up of one $Fe_3O_4@SiO_2@RF$ core@shell nanosphere with a diameter of 260 nm as a center, and four 150-nm PMO cubes as pods on the surface. By regulating the surface nucleation kinetic of PMO multi-site growth on polymeric RF shell, a family of multipods structured mesoporous nanocomposites with controllable surface topological structures are obtained (Janus nanostructure with only one PMO pod on RF surface, dual-pods and tri-pods with nearly plane distribution, three dimensional tetrahedral structured tetra-pods, etc.). The multipods topology endows the obtained mesoporous nanocomposites high bacteria adhesion efficiency through multivalent interactions. Due to the multipods structure enhanced bio-nano interactions and high surface area (~584 m² g⁻¹) for antibiotic loading, the tetra-pods nanoparticles perform nearly 100% bacteria segregation and over 90% long-term inhibition abilities.

## Results
### Tribulus-like mesoporous tetra-pods $Fe_3O_4@SiO_2@RF\&PMOs$.
Superparamagnetic $Fe_3O_4$ nanoparticles are prepared through a solvothermal method, which are successively coated with polymeric $SiO_2$ and RF through the sol–gel process, forming uniform $Fe_3O_4@SiO_2@RF$ core@shell structured nanospheres (Fig. 1a). Then, the mesoporous tetra-pods are fabricated through the surface-kinetics mediated multi-site nucleation and growth of PMOs on the surface of the $Fe_3O_4@SiO_2@RF$ nanoparticles: each $Fe_3O_4@SiO_2@RF$ core particle is surrounded by four separated PMO pods, forming a tribulus like tetrahedral structured tetra-pods $Fe_3O_4@SiO_2@RF\&PMOs$ mesoporous nanocomposite (denoted as T-RF&PMOs). As shown in transmission electron microscopy (TEM) images (Fig. 1b, and Supplementary Fig. 1), the magnetite $Fe_3O_4$ nanoparticles synthesized through the solvothermal method at 200 °C have a diameter of 100 nm, the thicknesses for the coated $SiO_2$ and RF layers are measured to be both 40 nm, together forming the $Fe_3O_4@SiO_2@RF$ core@shell structured nanospheres with 260 nm in diameter. For the T-RF&PMOs nanoparticles, TEM and scanning electron microscopy (SEM) images clearly demonstrate the 3D tetra-pods topological structure (Fig. 1b–d & Supplementary Fig. 2), and that the PMO nanocubes have similar size of ~150 nm in side length (Supplementary Fig. 3). The four PMO nanocubes are uniformly anchored on the surface of one $Fe_3O_4@SiO_2@RF$ nanosphere, and their relative position constitutes a nearly regular tetrahedron topology (Supplementary Fig. 4). Different elements in one $Fe_3O_4@SiO_2@RF$ nanosphere (iron, carbon) and four discrete PMO pods (silicon) can be clearly identified by the energy dispersive spectrometer (EDS) mapping (Fig. 1e), further demonstrating the tetra-pods topology. Three scattering peaks at q-value of 1.35, 1.51, and 1.65 nm⁻¹ are observed in the small-angle X-ray scattering patterns of the T-RF&PMOs nanoparticles, which correspond to the 200, 210, and 211 reflections of the cubic mesostructure (Pm$\bar{3}$n) (Supplementary Fig. 5). This result clearly indicates the ordered cubic mesostructure of the PMO cube pods, which is consistent with the high-resolution TEM (HRTEM) images of the PMO pods along the [210] and [001] zone axis (Fig. 1f, g). It's worth mentioning that the surface of the initial RF nanospheres without the deposition of PMO cubes is also covered by a very thin layer of organosilicas. The main difference between such layer and PMO cubes is that the thin layer of organosilica does not have ordered mesostructure. Nitrogen sorption isotherms of the obtained mesoporous T-RF&PMOs nanoparticles exhibit representative type-IV curve with a rapid increase of adsorption volume at a relative pressure of 0.2–0.4, indicating uniformity of the mesopores in the nanocomposites. The Brunauer −Emmett−Teller (BET) surface area is calculated to be as high as 584 m² g⁻¹ (Fig. 1h). From the pore size distribution curve obtained by the Barrett-Joyner-Halenda (BJH) method, the mesoporous Janus nanocomposites exhibit a narrow mesopore size distribution with an average diameter of ~2.6 nm, which is mainly derived from the ordered mesopores of the PMO pods. With the presence of the magnetic $Fe_3O_4$ cores, the T-RF&PMOs multipods nanoparticles exhibit superparamagnetic properties with a saturation magnetization value of 20 emu g⁻¹, which is enough for magnetic induced separations in the practical applications (Supplementary Fig. 6). The multipods nanoparticles are well dispersed in Luria-Bertani (LB) solution over 12 h, demonstrating a good stability (Supplementary Fig. 7). The stability of the multipods nanoparticles in LB, phosphate buffer saline (PBS), acid and basic solutions, etc. are also evaluated by dynamic light scattering (DLS). The results show that the average diameters of the nanoparticles remain nearly constant in different solutions over a long period of times, indicating no aggregation of the nanoparticles in the solutions (Supplementary Fig. 8). Altogether, the multipods

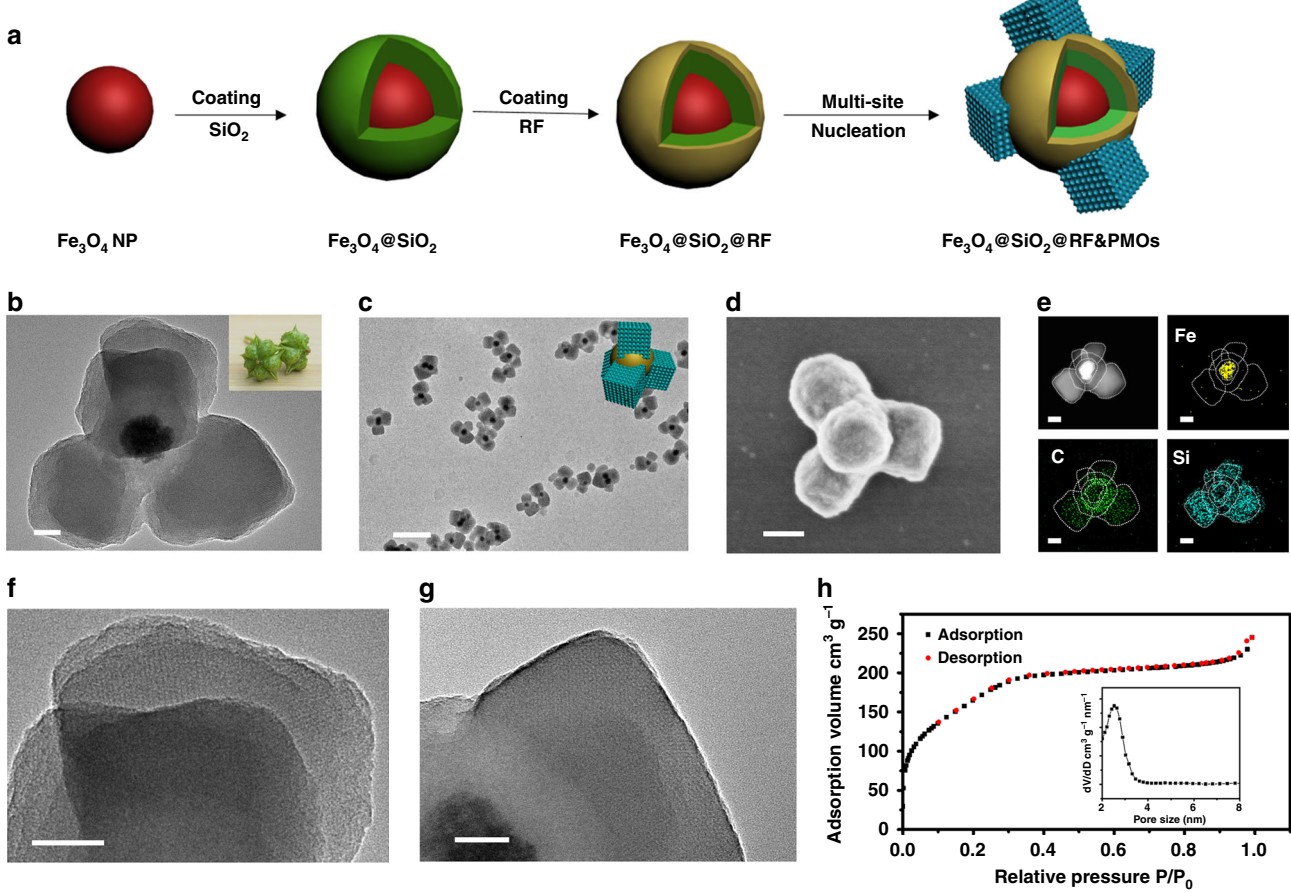

**Fig. 1** Synthesis and characterization of tribulus-like tetra-pods nanocomposites. **a** Fabrication process scheme of the tetra-pods Fe$_3$O$_4$@SiO$_2$@RF&PMOs nanoparticles based on the surface-kinetic mediated multi-site nucleation strategy. **b**, **c** TEM images with different magnifications, **d** SEM image and **e** element mapping of the tribulus-like tetra-pods Fe$_3$O$_4$@SiO$_2$@RF&PMOs mesoporous nanocomposites. Inset in **b**: a digital photo of tribulus seeds; Inset in **c**: 3D structural model of the tetra-pods Fe$_3$O$_4$@SiO$_2$@RF&PMOs nanoparticles. **f**, **g** HRTEM images of the typical PMO pods of tetra-pods Fe$_3$O$_4$@SiO$_2$@RF&PMOs nanoparticles along **f** [210] and **g** [001] zone axis. **h** The nitrogen sorption isotherms and pore size distribution (inset) of the tetra-pods nanoparticles. Scale bars represent 50 nm in **b**, **f**, **g**, 100 nm in **d**, **e**, and 500 nm in **c**. Source data underlying **h** are provided as a Source Data file

topology, high surface area and ordered mesostructure endow the T-RF&PMOs nanoparticles great potentials for applications with enhanced nano-bio interactions.

In stark contrast, the nucleation of cubic PMOs upon the bare Fe$_3$O$_4$@SiO$_2$ nanoparticles leads to the formation of Janus nanostructured Fe$_3$O$_4$@SiO$_2$&PMO with only one PMO pod (Fig. 2a)[28]. The Fe$_3$O$_4$@SiO$_2$ nanoparticles with a diameter of ~260 nm are the same size with Fe$_3$O$_4$@SiO$_2$@RF nanoparticles, only without the RF layer coating (all reaction conditions are the same) (Supplementary Fig. 9). It can be seen that the obtained nanoparticles show an obvious asymmetric Janus nanostructure, which are composed by closely connected Fe$_3$O$_4$@SiO$_2$ core@shell nanosphere (~260 nm in diameter) and a PMO nanocube (~ 250 nm in edge length). Compared with the PMO pods in the T-RF&PMOs, the size of the PMO cube in the Janus nanostructured is relatively larger. The cubic mesostructure (space group of Pm$\bar{3}$n), diameter of the mesopore etc. are maintained and consistent with that in the T-RF&PMOs. Fe$_3$O$_4$@RF nanoparticles with diameter of ~260 nm were also synthesized and used as the substrate for the multi-site nucleation and growth of PMOs. The mesoporous multipods Fe$_3$O$_4$@RF&PMOs can also be synthesized under the same reaction conditions (Supplementary Fig. 10). These results clearly indicate that the surface properties have significant impact on the multi-site nucleation of PMOs, which will be described in the following part.

**The controllable surface topological structures**. As mentioned above, the Janus/mono-pod and tetra-pods structured nanoparticles can be obtained by the heterogeneous nucleation and growth of PMO nanocubes on the SiO$_2$ and RF surface, respectively, indicating that the surface properties have significant impact on the multi-site nucleation of PMOs. However, when polyvinyl pyrrolidone (PVP) is used to modify the surface of Fe$_3$O$_4$@SiO$_2$@RF nanoparticles, the surface kinetics for PMOs growth is changed, the multi-site nucleation is suppressed, and the number of the PMO pods is restrained to only one, leading to reinstatement of the Janus Fe$_3$O$_4$@SiO$_2$@RF&PMO structures (Fig. 2c, Supplementary Fig. 11). The intermediate surface topological structure with 1–2 cubes grown on each nanoparticle is also obtained with the decreasing amount of PVP on the surface of Fe$_3$O$_4$@SiO$_2$@RF nanoparticles (Supplementary Fig. 12). Other kinds of polymers, such as poly(sodium-*p*-styrenesulfonate) (PSS) also possess such function to suppress multi-site nucleation of PMOs on RF surface (Supplementary Fig. 13). These results clearly indicate that the multi-site nucleation for the formation of the multipods architecture is mainly ascribed to the surface property of the RF outer layer, allowing precise manipulation of the multipods topological structure through particles' surface property manipulation.

To further investigate the effects of surface property on mediation of the multi-site nucleation for the formation of

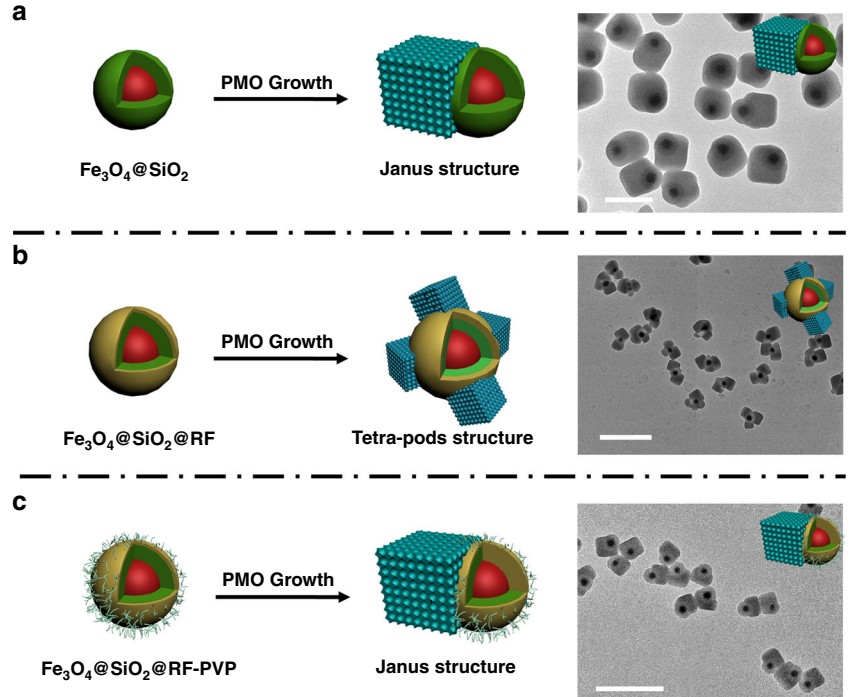

**Fig. 2** The growth of PMO cubes on nanoparticles with different surface compositions. **a** The growth of PMOs on the core@shell $Fe_3O_4$@$SiO_2$ nanospheres and the TEM image of the resultant Janus $Fe_3O_4$@$SiO_2$&PMO nanocomposites; **b** The multi-sites growth of PMOs on the $Fe_3O_4$@$SiO_2$@RF nanospheres and the TEM image of the resultant tetra-pods $Fe_3O_4$@$SiO_2$@RF&PMOs nanocomposites; **c** The growth of PMOs on the PVP modified $Fe_3O_4$@$SiO_2$@RF nanospheres and the TEM image of the resultant Janus $Fe_3O_4$@$SiO_2$@RF-PVP&PMO nanocomposites. Insets are the 3D structural model of the respective nanocomposites. Scale bars represent 500 nm in a, and 1 μm in **b**, **c**

multipods structures, the surface of the core@shell structured $Fe_3O_4$@$SiO_2$@RF nanoparticles is functionalized with phenolic amino (RF-A) or phenolic nitro (RF-N) groups, and the corresponding nanoparticles are denoted as $Fe_3O_4$@$SiO_2$@RF-A and $Fe_3O_4$@$SiO_2$@RF-N, respectively (Supplementary Fig. 14). After heterogeneous nucleation and growth of PMOs on the nanoparticles, the obtained $Fe_3O_4$@$SiO_2$@RF-A&PMOs multi-pods nanocomposites with amino groups mainly possess two cubic pods (Fig. 3a), less than that of tribulus-like $Fe_3O_4$@-$SiO_2$@RF&PMOs nanoparticles with tetra-pods topology (obtained by the heterogeneous nucleation and growth of PMOs on RF surface with phenolic hydroxyl group) (Fig. 3b). The side length of the cubes are ~260 nm, slightly larger than that of $Fe_3O_4$@$SiO_2$@RF&PMOs tetra-pods nanocomposites. In most $Fe_3O_4$@$SiO_2$@RF-A&PMOs dual-pods nanocomposites, the relative position among the two PMO cubes and $Fe_3O_4$@$SiO_2$@RF-A have a plane obtuse triangle structure or nearly linear alinement, indicating that the heterogeneous nucleation of PMOs on the polymeric RF-A surface tends to spontaneously keep away from each other to minimize the system energy. In contrast, the mesoporous $Fe_3O_4$@$SiO_2$@RF-N&PMOs nanocomposites with more than four PMO pods (6–8 cubes according to TEM images) can be obtained after heterogeneous nucleation and growth of PMOs on the RF-N surface with phenolic nitro functional groups (Fig. 3c). The PMO pods possess homogeneous size of ~80 nm, and are evenly distributed on the surface of the core–shell $Fe_3O_4$@$SiO_2$@RF-N nanoparticles. These results further imply that the localized surface property of the nanoparticles has great influence on the heterogeneous multi-sites nucleation of PMOs.

Theoretically, the number of nucleation sites is positively correlated with the surface area. To verify this assumption, core@shell $Fe_3O_4$@$SiO_2$@RF nanoparticles with the same surface property but different diameters are used for the heterogeneous

multi-sites nucleation and growth of PMOs. Uniform tetra-pods topologies can be formed based on $Fe_3O_4$@$SiO_2$@RF with a large diameter of 260 nm (Fig. 1). In contrast, tri-pods (Fig. 4a), bi-pods (Fig. 4b) and mono-pod (Fig. 4c) structured nanocomposites are obtained when the diameter of $Fe_3O_4$@$SiO_2$@RF nanoparticles decrease to 230, 200, and 160 nm, respectively. When the diameter of $Fe_3O_4$@$SiO_2$@RF nanoparticles increases to > 300 nm, more complicated multi-pods (> 10) nanostructures can be obtained (Supplementary Fig. 15). All the obtained multi-pods nanoparticles are uniform and well dispersed. The $Fe_3O_4$@$SiO_2$@RF nanoparticles are considered as an ideal sphere. The outer surface area of nanospheres can be calculated according to the equation $S = \pi d^2$ (d is the diameter of the nanoparticles). The correlation between the number of PMO cubes (nucleation site number) and the surface area per $Fe_3O_4$@$SiO_2$@RF nanoparticle is established, showing nearly linear relation (Fig. 4d). The average area for one nucleation-site on the pure RF surface is estimated to be about $6 \times 10^4$ nm$^2$. The linear correlation between the nucleation site number and surface area indicates that the formation of the multiple PMO nucleation sites on RF surface is an even distribution manner, which further induces the most stable and highest symmetric structure for the multipods nanocomposites. It is consistent with the nearly linear alinement for dual-pods nanocomposites and nearly regular tetrahedron structure for the tetra-pods nanocomposites. Beside the number of the PMO cubes, the average side length of the cubes decreased from 300 to 100 nm as increasing of the number of PMO cubes on the $Fe_3O_4$@$SiO_2$@RF surface from 1 to 8. When the total amount of the silane precursors is constant, the more cubes grown, the smaller the size of each one (Supplementary Fig. 16).

Besides the surface functional group and external surface area, the influence of surface structure (non-porous surface VS mesoporous surface) on the heterogeneous multi-site nucleation

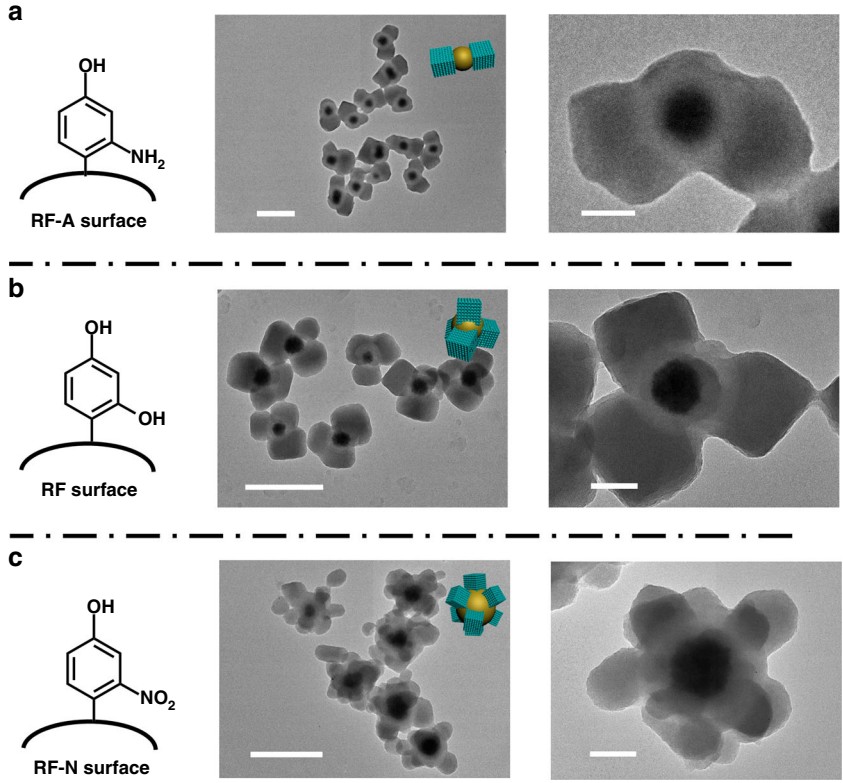

**Fig. 3** The growth of PMO cubes on nanoparticles with different functional groups. **a**–**c** The structure of the surface functional groups on Fe$_3$O$_4$@SiO$_2$@RF-A, Fe$_3$O$_4$@SiO$_2$@RF, and Fe$_3$O$_4$@SiO$_2$@RF-N nanoparticles and the corresponding TEM images with different magnifications of the obtained multipods nanocomposites: **a** The growth of PMO cubes on core@shell Fe$_3$O$_4$@SiO$_2$@RF-A nanoparticles with phenolic amino surface functional groups and TEM images with different magnifications of the obtained dual-pods Fe$_3$O$_4$@SiO$_2$@RF-A&PMOs nanocomposites; **b** The growth of PMO cubes on the core@shell Fe$_3$O$_4$@SiO$_2$@RF nanoparticles with phenolic hydroxyl surface functional groups and the TEM images with different magnifications of the obtained tetra-pods Fe$_3$O$_4$@SiO$_2$@RF&PMOs nanocomposites; **c** The growth of PMO cubes on the surface of Fe$_3$O$_4$@SiO$_2$@RF-N nanoparticles with phenolic nitro functional groups and TEM images with different magnifications of the obtained multipods Fe$_3$O$_4$@SiO$_2$@RF-A&PMOs nanocomposites. Insets are the 3D structural model of the respective nanocomposites. Scale bars represent 500 nm for images in the middle line and 100 nm for images in the right line

of PMO on RF surface was also investigated. The Fe$_3$O$_4$@mRF nanoparticles (mRF = mesoporous RF) with mesoporous RF shells are fabricated (Fig. 4e, g) and used for the multi-site nucleation and assembly of PMOs (Fig. 4f, h) on their surface. For the Fe$_3$O$_4$@mRF nanoparticles with a diameter of ~200 nm, four to six PMO cubes can be grown on each nanoparticles (Fig. 4f), which is more than the nuclei number on the above mentioned non-porous RF surface of the Fe$_3$O$_4$@SiO$_2$@RF nanoparticles (the same particle diameter) (Fig. 4b). When the size of the Fe$_3$O$_4$@mRF increases to 300 nm, the number of the PMO pods also increases to >10 (Fig. 4h, Supplementary Fig. 17). The increase of nuclei number for the multi-site nucleation of PMO on mesoporous RF surface indicates that the mesopore channels can provide additional active surface area for the multi-site nucleation of PMO, leading to increased pod numbers.

The surface-kinetics mediated multi-site nucleation strategy is universal in manipulating the surface topological structure of various nanomaterials, including Fe$_2$O$_3$ spindles, Fe$_2$O$_3$ cubes, graphene oxide (GO) and carbon nanotubes (CNT). The surfaces of these materials are first modified with RF layers (Supplementary Fig. 18), then PMO cubes can be grown through the surface-kinetics mediated multi-site nucleation strategy to form multipods surface topological structures (Supplementary Fig. 19). However, unlike the uniform multi-site nucleation of PMOs on isotropic Fe$_3$O$_4$@SiO$_2$@RF nanospheres, the distribution of the PMO cubes on the spindles, cubes and nanotubes are not as uniform as the tetra-pods Fe$_3$O$_4$@SiO$_2$@RF&PMOs

nanoparticles, which can be attributed to the anisotropic morphology of the spindles, cubes and nanotubes. Such general applicability promises the fabrication of multipods nanocomposites with various functions. Furthermore, this surface-kinetics mediated multi-site nucleation strategy can also be utilized for multi-site growth of PMO nanorods with hexagonal mesostructure (space group of p6mm), resulting in multipods morphology with multiple nanorod as pods, indicating that it is generalized to other kinds of mesoporous structures (Supplementary Fig. 20). Altogether, such method exhibits both general applicability as well as precise manipulation, showing a potential in manufacturing nanomaterials' surface topologies.

**Enhanced bacterial adhesion and inhibition.** Bacterial infection presents a major threat to public health, especially with the increasing level of antimicrobial resistance in recent decades, causing large number of illness and death[33,34]. Present methods for elimination of bacteria all call for strong interactions between materials and bacteria. In analogy of tribulus' adhesion toward animal's skin and fur, it is hypothesized that the fabricated T-RF&PMOs with multipods topology can also provide enhanced interaction toward bacteria. *Escherichia coli* (*E. coli*), a typical Gram-negative strain[35,36], is used as a model to evaluate the bacterial adhesion ability of T-RF&PMOs nanoparticles with tetra-pods topological structure. The asymmetric J-RF&PMO with Janus morphology and spherical core@shell structured Fe$_3$O$_4$@SiO$_2$@RF@PMO (denoted as RF@PMO) nanoparticles

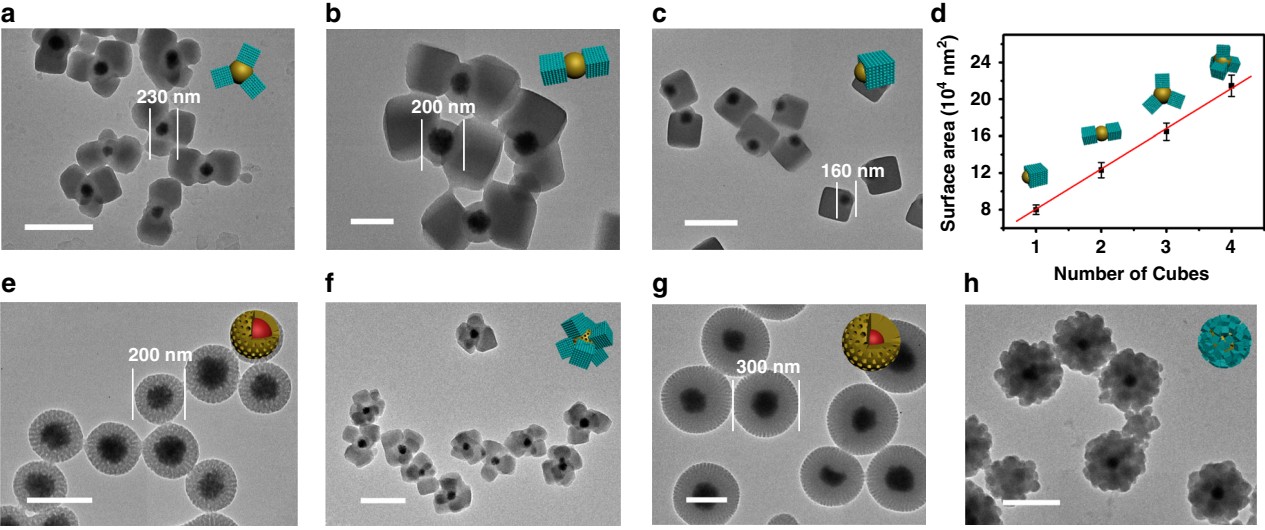

**Fig. 4** The growth of PMO cubes on nanoparticles with different outer surface area. **a–c** TEM images of the multipods Fe₃O₄@SiO₂@RF&PMOs nanocomposites with controllable number of mesoporous PMO cubes by tuning the diameter of Fe₃O₄@SiO₂@RF substrate nanoparticles: **a** 230 nm, **b** 200 nm and **c** 160 nm; **d** The linear fitting of the number of PMO cubes of the multipods Fe₃O₄@SiO₂@RF&PMOs nanoparticles versus single particle surface area of the initial Fe₃O₄@SiO₂@RF nanoparticles; **e–h** TEM images of the Fe₃O₄@mRF nanoparticles with different diameters (**e**, 200 nm; **g**, 300 nm) and the obtained corresponding multipods structured Fe₃O₄@mRF&PMOs nanoparticles **f**, **h**. Insets are 3D structural models of the multi-pods Fe₃O₄@SiO₂@RF&PMOs nanoparticles, Fe₃O₄@mRF nanoparticles and multi-pods Fe₃O₄@mRF&PMOs nanoparticles. Scale bars represent 200 nm in **b**, **e**, **g** and 500 nm in **a**, **c**, **f**, **h**. The bars represent mean ± s.d. derived from n = 10 randomly selected nanospheres. Source data underlying **d** are provided as a Source Data file

(Supplementary Fig. 21) are set as control groups to investigate the influence of nanoparticles' surface topological structure on nano-bio interactions.

After incubating *E. coli* with the fabricated nanoparticles with different surface topologies (T-RF&PMOs, J-RF&PMO, RF@PMO), the SEM image clearly demonstrates that large amount of T-RF&PMOs nanoparticles are adhered on the surface of the rod-like bacteria (Fig. 5a, Supplementary Fig. 22). The TEM image shows that the four pods of the T-RF&PMOs nanoparticles grabbed the bacteria's surface according to the multivalent interactions, just like tribulus attach to animals' fur (Fig. 5b). In contrast, J-RF&PMO and RF@PMO nanoparticles without the multipods structure are barely observed in the SEM and TEM images (Supplementary Fig. 23, 24), indicating that they cannot be efficiently adhered onto *E. coli*'s surfaces. Quantitative analysis results by inductively coupled plasma (ICP) show that the adhesion amount of the multipods T-RF&PMOs is almost fourfold comparing to the control groups (Fig. 5c), demonstrating the significance of the multipods topological structure for the enhanced nano-bio interaction. It should be noted that, the concept multivalent here is different from the multivalent that usually refers to increasing the strength of antibodies' binding towards targets with more than one binding domains according to chemical bonding. The multivalent in this article refers to the multipods nanoparticles interacting with biological hosts through the multiple pods, which is mainly based on a Van der Waals' interaction, and affected by the morphology of the nanoparticles.

Taking the advantage of the superparamagnetic Fe₃O₄ nanoparticles in the T-RF&PMOs multipods nanocomposites, it can be used for the magnetic field induced elimination of T-RF&PMOs adhered *E. coli* in water (Fig. 5d). Since a large amount of nanoparticles are adhered on *E. coli*'s surface, the bacteria are easily separated along by the magnet under a magnetic field. The nearly transparent culture solution can be obtained after magnetic induced separation, indicating the high bacterial segregation efficiency (Supplementary Fig. 25). The elimination efficiency of the bacteria with the tetra-pods structured T-RF&PMOs nanocomposites is almost 100%,

which is more than three and five times higher than that of the J-RF&PMO with Janus morphology or RF@PMO with a core@shell structure (Fig. 5e).

Many antimicrobial drugs, lysozyme for example, are insensitive toward Gram-negative strains such as *E. coli*, because of the protection of bacteria's outer membrane[37,38]. Though many efforts have been made to fabricate nanomaterials as carriers of lysozyme for increasing its antimicrobial efficiency[39–43], the drug carriers often suffer from the poor bacteria adhesion ability. Herein, the T-RF&PMOs nanoparticles with both high-surface-area mesopores for drug loading and tribulus-like tetra-pods structure for enhanced bacteria adhesion, are employed to antibacterial assessments (Supplementary Fig. 26). The surface topological structure and good dispersity of T-RF&PMOs is maintained after lysozyme loading (Supplementary Fig. 27). The results show that the antibacterial activity of lysozyme-immobilized T-RF&PMOs with tetra-pods topology is over 90% at the lysozyme concentration of ~400 µg mL⁻¹ (Fig. 5f). Such bacteria inhibition ability exceeds the control groups without multipods topologies. The long-term bacterial inhibition property is also evaluated to further demonstrate the antibacterial ability of lysozyme loaded tribulus-like T-RF&PMOs nanoparticles. The lysozyme-immobilized T-RF&PMOs nanoparticles can retain over 90% bacterial inhibition efficiency with negligible *E. coli* growth even in a long term of 3 days (Fig. 5g). In stark contrast, bacterial inhibition efficiency for the free lysozyme, lysozyme loaded J-RF&PMO and RF@PMO nanoparticles are much lower (<50%). The colony forming units (CFU) on agar plates of treated bacteria solution clearly show that the number of colonies in T-RF&PMOs treated groups is significantly lower compared with the other control groups, further evidencing the multipods enhanced bacterial adhesion and inhibition ability (Supplementary Fig. 28).

## Discussion
The surface-kinetics mediated multi-site nucleation strategy is proposed for the fabrication of the above mentioned mesoporous

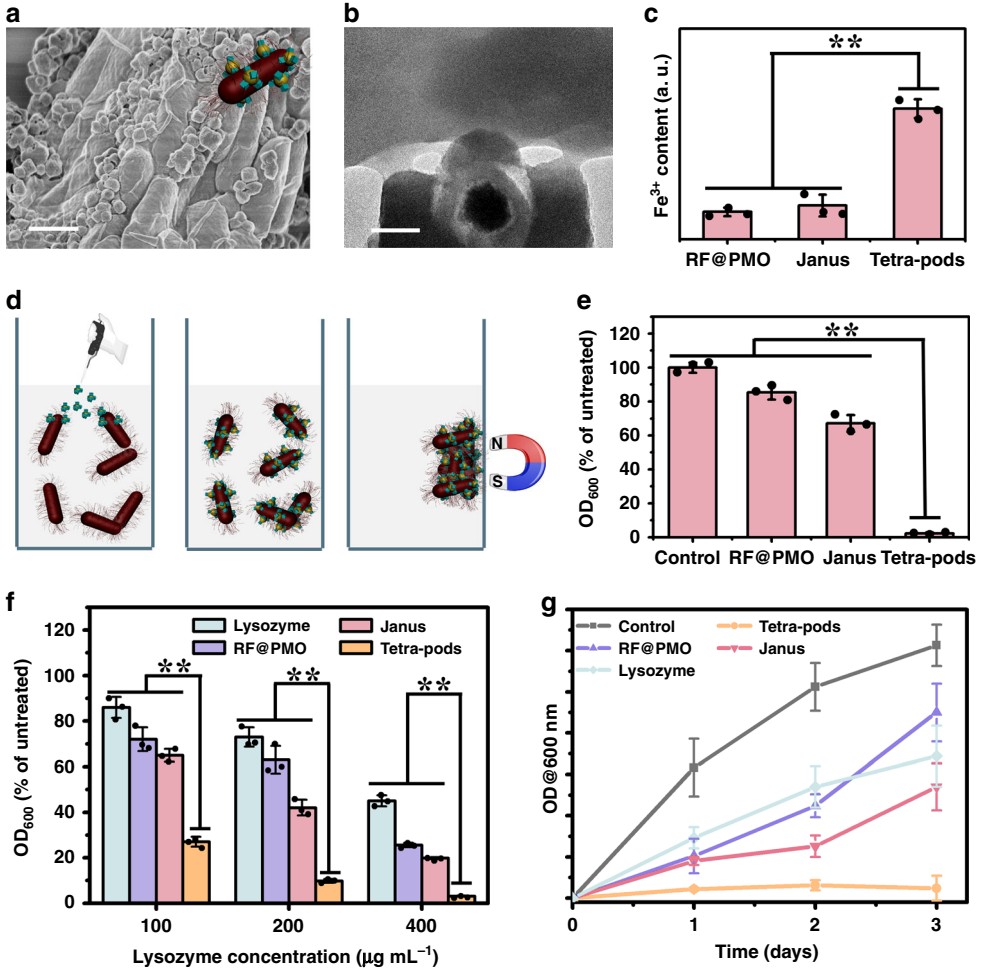

**Fig. 5** Surface topologies enhanced bacterial adhesion and inhibition. **a** SEM, **b** TEM images and (inset of a) 3D structural model of the tetra-pods $Fe_3O_4@SiO_2@RF\&PMOs$ nanoparticles adhered on *E. coli* surfaces. Scale bars represent 1 μm for a, 100 nm for **b**. **c** Quantitative analyses of the nanoparticles adhered on bacteria from ICP based on $Fe^{3+}$ concentration: the tetra-pods $Fe_3O_4@SiO_2@RF\&PMOs$ (Tetra-pods), Janus $Fe_3O_4@SiO_2@RF\&PMO$ (Janus) and core@shell $Fe_3O_4@SiO_2@RF\&PMO$ (RF@PMO) nanocomposites. **d** Schematic illustration of the adhesion and magnetic induced separation of *E. coli* with the multipods nanocomposites. **e** Quantitative analysis of *E. coli* in the PBS solution after the magnetic induced separation process. Optical density (OD) value is positive correlated with the amount of bacteria in the solution. **f** *E. coli* inhibition efficiency of different nanoparticles loaded with the same amount of lysozyme within 24 h. **g** The time-dependent *E. coli* inhibition evaluation of different nanocomposites loaded with the same amount of lysozyme (400 μg mL$^{-1}$). The bars represent mean ± s.d. derived from $n = 3$ groups of bacteria suspension. Source data underlying **c**, **e**–**g** are provided as a Source Data file. The statistical analysis was performed using one-way analysis of variance (ANOVA), followed by post hoc Tukey's method. **$P < 0.01$

multipods with precisely tunable surface topological structures (Fig. 6a). Prior to the heterogeneous nucleation on solid surface, the concentration of surfactants/silicate oligomers in the solution increases with the hydrolysis of silanes as the silica precursors (Fig. 6b), and the generation speed of the oligomers in the solution is decreased as the prolonging of hydrolysis of the silane. When the concentration of the surfactants/silicate oligomers rises to the critical nucleation concentration ($C_n$), large amount of oligomers in the solution are instantly consumed for the formation of nucleation sites on the solid surface. When this consumption speed for the nucleation on solid surface is faster than the generation speed of the oligomers from silanes, it can result in decrease of the oligomer concentration in the solution. Until the oligomer amount in the solution is lower below the critical nucleation concentration ($C_n$), the heterogeneous nucleation is terminated and the PMO nuclei continuously grow into larger cubic domains on solid surface. In the whole process, the duration time ($\Delta t_n$) that the oligomer amount in solution is above the critical nucleation concentration is crucial for the determination

of the nucleation number on solid surface. The longer the duration time of nucleation ($\Delta t_n$) is, the more nucleation sites are formed. Traditionally, the reaction kinetics in the solution for the generation of oligomers is used to manipulate this duration time of nucleation ($\Delta t_n$)[44–46]. Herein, the surface nucleation kinetics of oligomers on the solid surface, which determines the critical nucleation concentration ($C_n$), is proposed and used for the manipulation the duration time of nucleation ($\Delta t_n$).

Since oxygen anions' nucleophilicity on the surface of polymeric RF is stronger than that of oxygen in Si-OH groups on $SiO_2$ surface (Fig. 6b), the reaction activity of silane nucleophilic substitution (i.e., nucleation capability) on RF surface is higher than that on $SiO_2$ surface, meaning that the critical nucleation concentration of surfactants/silicate oligomers on RF surface ($C_1$) is lower than that on $SiO_2$ surface ($C_2$) (Fig. 6a)[47,48]. So, when all the experimental conditions are consistent, the starting time of the heterogeneous nucleation on the RF surface ($T_1$) is earlier than that on $SiO_2$ surface ($T_2$), indicating that the generation speed of the oligomers in the solution is quicker at the relatively

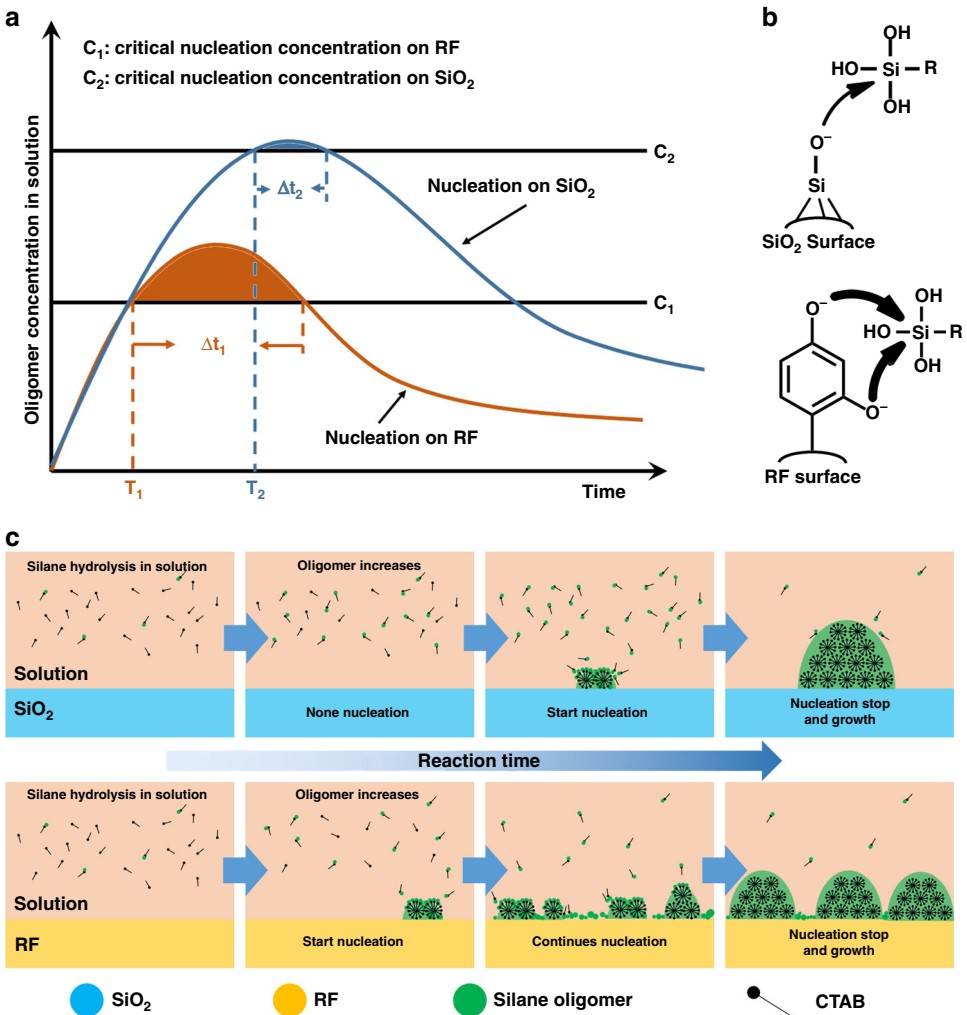

**Fig. 6** The mechanism of the surface-kinetics mediated multi-site nucleation strategy. **a** The schematic illustration of the surfactants/silicate oligomers concentration in the solution at different reaction time with the presence of different nucleation substrates ($SiO_2$ and RF). $C_n$ is the critical nucleation concentration of surfactants/silicate oligomers for the heterogeneous nucleation of PMOs on the particle surface; $T_n$ and $\Delta t_n$ are the starting and duration time of the heterogeneous nucleation on the particle surface. **b** The schematic illustration of the reaction activity of the silane nucleophilic substitution on $SiO_2$ and RF surfaces (i.e., nucleation activity). Because the reaction activity of silane nucleophilic substitution on RF surface (with abundant Ph-OH group) is higher than that on $SiO_2$ surface (with abundant Si-OH group), in Fig. 6a, the critical nucleation concentration of surfactants/silicate monomer on RF surface ($C_1$) is lower than that on $SiO_2$ surface ($C_2$). **c** Schematic illustration of the surface-kinetics mediated multi-site nucleation of surfactants/silicate oligomers on $SiO_2$ and RF surface. The prolonged duration time of PMO nucleation on RF surface comparing to that of on $SiO_2$ surface induces the continuous multiple nucleation of surfactants/ silicate oligomers on the surface of $Fe_3O_4$@$SiO_2$@RF nanoparticles, and thus the multipods topological structure

earlier starting time of the nucleation on the RF surface. In comparison, the generation speed of the oligomers is slower at the relatively later starting time of the nucleation on the $SiO_2$ surface, because the generation speed in the solution is decreased as the prolonging of hydrolysis of the silane precursors, which can be confirmed by the experimental results (Supplementary Fig. 29, 30). Due to the difference on the generation speed of the oligomers at their respective starting time of nucleation induced by the different surface reaction kinetics, the duration time of the nucleation on RF surface ($\Delta t_1$) is longer than that on $SiO_2$ surface ($\Delta t_2$) (Fig. 6b), inducing the continuous multiple nucleation of surfactants/silicate oligomers on the RF surface and thus the multipods topological structure (Fig. 6c).

The surface kinetics of the heterogeneous nucleation of surfactants/silicate oligomers can be precisely manipulated through changing the surface functional groups of RF layers (Fig. 3). Amino is a weaker nucleophilic group compared to phenolic hydroxyl groups on RF surface, while phenolic nitro group is stronger due to

strong electron-withdrawing ability of $NO_2$ groups. Thus, the critical nucleation concentration of the surfactants/silicate monomers on RF-N surface with phenolic nitro groups is the lowest, and that on RF-A surface with phenolic amino groups is the highest among the three (Supplementary Fig. 31), resulting in differed starting and duration time of the nucleation, thus different number of nucleation sites. These inferences are consistent with the experimental results (Fig. 3). The surface kinetics of the heterogeneous nucleation of surfactants/silicate oligomers on RF can also be precisely manipulated by the surface modification with polymers (PVP, Fig. 2c). The nucleophilic ability of carbonyl group in PVP is weaker than phenolic hydroxyl groups in RF. After the surface modification with PVP, the initial phenolic hydroxyl groups in RF is covered by carbonyl groups with weaker nucleophilic ability. Therefore, the surface kinetics of RF is decreased by PVP coverage. The critical nucleation concentration of oligomers on RF-PVP surface is increased. Thus, the starting time of the heterogeneous nucleation is delayed. Duration time of nucleation is shortened and

the multi-site nucleation of PMOs is hindered, resulting in Janus architectures.

In comparison, although the generation speed of silicate oligomers can be increased by increasing the amount of the silane precursors and the alkalinity of the solution, such increase is invalid for the heterogeneous multi-site nucleation. The high concentration of the silane oligomers in the solution induced by increasing the precursor and alkalinity of the solution, can only result in the homogeneous nucleation of the oligomers in the solution, rather than the heterogeneous nucleation on the RF surface (Supplementary Fig. 32).

This surface-kinetics mediated multi-site nucleation strategy can also be utilized to demonstrate the correlation between the number of nucleation sites and particle diameter (Fig. 4d). When the reaction condition is retained constant, the surface reaction kinetics of silane condensation on $Fe_3O_4@SiO_2@RF$ nanoparticles is fixed, as well as the number of PMOs nucleation sites per unit area. However, as the number of nucleation sites on each particle is determined by its outer surface area: the larger the nanoparticles are, the larger surface area for nucleation is, the higher the number of nucleation sites can be. So, the number of PMO pods on the surface can be precisely controlled by the diameter of $Fe_3O_4@SiO_2@RF$ nanoparticles. In the case of $Fe_3O_4@mRF$ nanoparticles (Fig. 4e–h), mesopores endow the RF surface with higher surface area, and thus higher PMO nucleation site number as well.

In summary, the surface-kinetics mediated multi-site nucleation strategy is developed for the fabrication of a family of uniform mesoporous multipods $Fe_3O_4@SiO_2@RF\&PMOs$ nanocomposites with controllable surface topologies. The mesoporous multipods structures contain one core@shell $Fe_3O_4@SiO_2@RF$ nanosphere as a center and controllable number of multiple PMO cubes circling the sphere as pods. The number of nucleation sites can be precisely controlled from one (Janus nanostructure) to four (tetra-pods nanostructure) by mediation of the surface properties of RF. Due to the multipods structure enhanced nano-bio interaction, the tribulus-like tetra-pods mesoporous nanocomposites possess high bacteria adhesion capacity and perform nearly 100% bacterial segregation as well as long-term inhibition over 90%. We consider that the developed surface-kinetics mediated multi-site nucleation strategy provides a pathway in the rational design of functional mesoporous nanoparticles with controllable surface topological structures for enhanced nano-bio interactions.

## Methods

**Synthesis of $Fe_3O_4$ nanoparticles**. Magnetic $Fe_3O_4$ nanoparticles with an average diameter of 120 nm were synthesized as following[49]. $FeCl_3·6H_2O$ (3.25 g, 20 mmol), ethylene glycol (100 mL), trisodium citrate (1.30 g, 4.4 mmol) and NaAc (6.00 g, 44 mmol) were mixed and sonicated for 1 h. The mixture was then transferred into a 200-mL Teflon-lined stainless-steel autoclave. The autoclave was heated at 200 °C for 10 h. After cooling down to room temperature, the products were washed with distilled water and ethanol and finally dispersed in ethanol for further usages.

**Synthesis of core@shell $Fe_3O_4@SiO_2$ nanoparticles**. Core@shell structured $Fe_3O_4@SiO_2$ nanoparticles with a diameter of 180 nm were synthesized as following[50]. Typically, 40 mg of the $Fe_3O_4$ nanoparticles obtained above were dispersed in the mixture of ethanol (35 mL) and $H_2O$ (8.75 mL). Then, ammonium hydroxide (25 wt%, 0.75 mL) was added with stirring. TEOS (75 μL) was then added dropwise. The mixture was stirred for 12 h at room temperature, then centrifuged and washed with water and ethanol. Finally, the core@shell $Fe_3O_4@SiO_2$ nanoparticles with a diameter of 180 nm were obtained. For the synthesis of the $Fe_3O_4@SiO_2$ nanoparticles with a diameter of 250 nm, the amount of TEOS was increased to 200 μL and the other conditions were retained unchanged.

**Synthesis of $Fe_3O_4@SiO_2@RF$ nanoparticles**. The coating of RF layers on the core@shell $Fe_3O_4@SiO_2$ nanoparticles to form $Fe_3O_4@SiO_2@RF$ core@shell@shell nanospheres with a diameter of ~260 nm was realized through an extended Stöber method[44]. In a typical synthesis, 60 mg of the core@shell structured $Fe_3O_4@SiO_2$ nanoparticles was dispersed in a mixture of $H_2O$ (120 mL), ethanol (240 mL), resorcinol (500 mg, 4.5 mmol) and formaldehyde (500 μL, 16.7 mmol). 6.0 mL of ammonium hydroxide (25 wt%) was added dropwise to initialize the reaction. The

mixture was stirred for 5 h at room temperature, then the product was centrifuged and washed with water and ethanol. For the preparation of $Fe_3O_4@SiO_2@RF$ nanoparticles with a diameter of 160, 200, 230 nm, the reaction time was shortened from 5 to 1.5, 3, 4 h, respectively.

**Synthesis of tetra-pods $Fe_3O_4@SiO_2@RF\&PMOs$ nanoparticles**. The multi-site nucleation of highly ordered PMOs on the core@shell structured $Fe_3O_4@SiO_2@RF$ nanoparticles was carried out through the surface-kinetics mediated multi-site nucleation strategy. In a typical synthesis, 10 mg of the above $Fe_3O_4@SiO_2@RF$ nanoparticles with a diameter of ~ 260 nm was dispersed in 38 mL of $H_2O$ and 2.0 mL of ethanol. In total 150 mg of cetyltrimethyl ammonium bromide (CTAB) and 1.8 mL of ammonium hydroxide (25 wt%) were added in succession. The mixture was stirred for 30 min, then 50 μL of bis(triethoxysilyl)ethane (BTEE) was added dropwise with stirring for another 3 h. The product was centrifuged and washed with water and ethanol.

**Bacteria adhesion**. *E. coli* (American type culture collection No. 12435) was first cultured in Luria–Bertani (LB) liquid medium (tryptone $10\,g\,L^{-1}$, yeast extract $5.0\,g\,L^{-1}$, NaCl $10\,g\,L^{-1}$, pH 7.0) at 37 °C under shaking at 200 rpm for 12 h. After diluting the bacterial suspension with Luria-Bertani (LB) medium, the nanoparticles dispersed in phosphate buffer saline (PBS) solution ($1.0\,mg\,mL^{-1}$) were added. The mixture was further incubated in 37 °C for 24 h. The samples were then fixed and stained with 2.5% glutaraldehyde. A drop of bacteria suspension was dropped and dried on the cupper grid for TEM characterization. The dried samples were sputter-coated with gold for SEM imaging. More than 10 positions were randomly selected to take the electron microscope images. The quantitative analyses of the nanoparticles adhered on bacteria were determined by inductively coupled plasma (ICP) based on $Fe^{3+}$ concentration.

**Reporting Summary**. Further information on research design is available in the Nature Research Reporting Summary linked to this article.

## Data availability
Data supporting the findings of this study are available within the article and the associated Supplementary Information Section. Any other data are available from the corresponding authors upon reasonable request. The source data underlying Figs. 4d, 5c, 5e–5g and Supplementary Figs. 4, 8, 16 and 26 are provided as a Source Data file.

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

## Acknowledgements

The work was supported by the National Key R&D Program of China (2017YFA0207303, 2018YFA0209400), Key Basic Research Program of Science and Technology Commission of Shanghai Municipality (17JC1400100), National Natural Science Foundation of China (21875043, 21733003, 21701027), Natural Science Foundation of Shanghai (18ZR1404600), and Shanghai Sailing Program (17YF1401000). This work was funded by the Deanship of Scientific Research at Princess Nourah Bint Abdulrahman University, through the Research Groups Program Grant No. RGP-1438-0006.

## Author contributions

D.Z., X.L., and T.Z. contributed to the conception and design of the experiments, analysis of the data and writing the manuscript. L.C., P.W., B.L., R.L., F.Z., and X.L. assisted T.Z. for the synthesis of materials and the data collection and analysis, A.A.K. and W.N.H. involved in partial bacteria data and analysis. All authors contributed to the discussion and manuscript preparation.

## Competing interests

The authors declare no competing interests.
