## [Peer Review File · Nature Communications]

Reviewers' comments:

Reviewer #1 (Remarks to the Author):

This manuscript proposed a surface-kinetics mediated multi-site nucleation strategy for the fabrication of unique mesoporous multipods with precisely tunable surface topological structures. A series of multipods mesoporous nanocomposites (such as Janus with only one pod, dual-pod, tri-pods, and three-dimensional tetrahedral structured tetra-pod) are obtained via manipulating the number of nucleation sites through mediating surface kinetics. The mesoporous nanoparticles show outstanding bacteria segregation and long-term inhibition after antibiotic loading. This manuscript is written well and the research outcome is broad interesting for the readers. I suggest that this manuscript can be accept for publication after my comments are addressed. Comments:

1. The authors succeeded in synthesizing the Fe₃O₄@SiO₂@RF nanoparticle with three morphologies. However, in page 10, products with different morphologies can also be obtained by controlling the size of Fe₃O₄@mRF. What is the role of SiO₂ in this material?
2. When the PVP is used to modify the surface of Fe₃O₄@SiO₂@RF nanoparticles, a special Janus structure is obtained. The multi-site nucleation is suppressed by adding PVP. How does the PVP change the surface kinetics of the material? Whether the mechanism of PVP, amine group and nitro group regulating surface property is the same? Please discuss more about this.
3. The linear correlation between the nucleation site number and surface area is given in the paper. How to understand the surface area and how to calculate it in this paper? The author said the mesopore channels can provide additional "active" surface area for the multi-site nucleation. Dose the author consider the surface area provided by the mesopores when making the linear curve between the nucleation site number and surface area?
4. The author reported that the generation speed of the oligomers in the solution is quicker at the relatively earlier starting time of the nucleation on the RF surface than that of SiO₂ surface. Are there any characterizations and evidence for that?
5. Check the whole manuscript carefully. There are some mistakes, such as ".....(c) 160 nm; (c) The linear fitting of the number of PMO cubes....." in the caption of Figure 4.

Reviewer #2 (Remarks to the Author):

This manuscript is interesting which demonstrates the controllable fabrication of multipods mesoporous Fe₃O₄@SiO₂@RF&PMOs nanocomposites with precisely controlled surface topological structures. The obtained unique nanostructures and proposed surface kinetics mediated multi-site nucleation strategy are intriguing. The findings in this manuscript are crucial for rational design of mesoporous nanoparticles with controllable topological structure and bio-applications with enhanced nano-bio interactions. Overall, the work has been carefully performed. I recommend acceptance of this manuscript after the following comments are addressed.

1. Although the authors demonstrated the general applicability of the surface kinetics mediated multisite nucleation strategy for forming multipod surface topologies on various kinds of nanomaterials, it seems that ordered surface topology is achieved only on nanospheres (in the text of the manuscript), whereas for spindles, cubes and nanotubes (Figure S14), the grown pods are randomly distributed. The authors should give some discussion on this.
2. For the surface of the initial RF nanospheres without the PMO cube nucleation and growth, is the RF surface exposed or covered by organosilica? More discussion on this point should be provided in the revised manuscript.
3. The linear correlation between nucleation site number and particle size is well established in this manuscript. From the TEM images, it seems that the size of the PMO cubes is also quite related with the number of the cubes, which should be clarified.
4. The authors claimed that the number of nucleation site is related to the speed of oligomer consumption and generation in the solution, which can be tunned by manipulating the surface property of the nanoparticles for the nucleation. Theoretically, increasing the added amount of

silane precursor can also increase the generation speed of silane oligomer, therefore affecting the number of nucleation site. So, I suggest the authors to provide some experiments to further support the proposed mechanism.

5. Some other minor points: (1) The concept of "multivalent interaction" is also used in design of drug molecule/antibody. The difference between this "multivalent" and the multivalent of nanoparticles should be provided in this article. (2) In Figure 6b, the surface kinetics is attributed to the difference in nucleophilicity between Si-OH groups and oxygen anions of RF. I suggest using wider arrow to represent stronger nucleophilicity. The present illustration may be misleading, suggesting that the main difference is the OH group number. (3) In Figure 5, the significant difference value needs to be provided.

Reviewer #3 (Remarks to the Author):

This manuscript focused on the use of surface-kinetics mediated multi-site nucleation strategy to synthesize mesoporous multipods with different topological structures. The authors successfully demonstrated their ability to tune the surface topology of nanoparticles. I do not have any major concerns on the novelty with respect to the fabrication. Relatively speaking, a weakness of this paper is on the demonstration of potential applications (i.e., the section on bacterial adhesion and inhibition).

Methodological details in collecting the TEM/SEM images – The authors are encouraged to 1) demonstrate representativeness of the images presented and 2) further explain the experimental details on how the collection of these images was conducted. For examples: How many images were taken per sample? How much variation was observed between images? In other words, it's important to demonstrate that effort has been taken to minimize subjective bias in imaging the nanoparticles.

Interpretation of TEM/SEM images – The authors are encouraged to exercise caution when interpreting these images. For examples: 1) on Page 12, the authors claimed that the nanoparticles "have weak interactions with the bacteria" and 2) on Page 14, "the anchoring strength of T-RF&PMOs nanoparticles on the bacteria is very strong and stable". TEM/SEM does not provide information about binding affinity. Using words like "strong", "weak", and "strength" is an over-interpretation of the image.

E. Coli incubation – It seems that the incubation of E. coli with the nanoparticles was done at high ionic strength (based on the content of LB medium provided in S.I.). Did these nanoparticles aggregate in such condition? If so, were the aggregation dynamics (extent and kinetics) different for the various types of nanoparticles? And how would aggregation affect their reactivity with E. Coli? If there was no aggregation, what enables the stability of the nanoparticles?

Manuscript Title: Surface-Kinetics Mediated Unique Mesoporous Multipods for Enhanced Bacterial Adhesion and Inhibition

Point-by-Point Response to Referees

Reviewer #1:

This manuscript proposed a surface-kinetics mediated multi-site nucleation strategy for the fabrication of unique mesoporous multipods with precisely tunable surface topological structures. A series of multipods mesoporous nanocomposites (such as Janus with only one pod, dual-pod, tri-pods, and three-dimensional tetrahedral structured tetra-pod) are obtained via manipulating the number of nucleation sites through mediating surface kinetics. The mesoporous nanoparticles show outstanding bacteria segregation and long-term inhibition after antibiotic loading. This manuscript is written well and the research outcome is broad interesting for the readers. I suggest that this manuscript can be accept for publication after my comments are addressed.

Response: We thank the reviewer very much for the positive comments.

Comments:

1. The authors succeeded in synthesizing the $Fe_3O_4@SiO_2@RF$ nanoparticle with three morphologies. However, in page 10, products with different morphologies can also be obtained by controlling the size of $Fe_3O_4@mRF$. What is the role of SiO_2 in this material?

Response: We appreciate the reviewer's comments. The multisite nucleation of periodic mesoporous organosilica (PMOs) and the final morphology of the mesoporous multipods nanocomposites is only affected by the surface property of the outer resorcinol-formaldehyde (RF) resin layer of the $Fe_3O_4@SiO_2@RF$ nanoparticles. The SiO_2 middle layer of the $Fe_3O_4@SiO_2@RF$ nanoparticles does not affect the surface topological structure of the final nanocomposites after the multi-site nucleation and growth of PMOs. The reason for adding such a middle layer is that the mono-dispersity of the obtained $Fe_3O_4@RF$ nanoparticles is not as good as $Fe_3O_4@SiO_2@RF$ nanoparticles. Therefore, the $Fe_3O_4@SiO_2@RF$ nanoparticles are used as the typical model sample to investigate the multi-site nucleation of PMOs on RF surface in this manuscript.

In the revised manuscript, the $Fe_3O_4@RF$ nanoparticles with the same diameter of $Fe_3O_4@SiO_2@RF$ nanoparticles were also fabricated (**Supplementary Figure S10**). The multisite nucleation of PMOs on $Fe_3O_4@RF$ nanoparticles is identical to that on the $Fe_3O_4@SiO_2@RF$ nanoparticles. The mesoporous multipods $Fe_3O_4@RF$ &PMOs can also be synthesized based on the surface-kinetics mediated multi-site nucleation strategy, evidencing that SiO_2 has no effect on the multi-site nucleation and growth of PMOs.

Accordingly, we have added the following illustration on Page 6 in the revised manuscript:

Fe₃O₄@RF nanoparticles with diameter of ~ 260 nm were also synthesized and used as the substrate for the multi-site nucleation and growth of PMOs. The mesoporous multipods Fe₃O₄@RF&PMOs can also be synthesized under the same reaction conditions (**Supplementary Figure S10**).

We have also added Supplementary Figure S10 in Page S11 of the revised Supporting Information:

Supplementary Figure S10. TEM images of the core@shell Fe₃O₄@RF nanoparticles (A) with a diameter of 260 nm prepared by a modified Stöber method³ and the multipods Fe₃O₄@RF&PMOs nanocomposites (B) prepared by the surface-kinetics mediated multi-site nucleation strategy. At the same reaction conditions, the obtained multipods Fe₃O₄@RF&PMOs nanocomposites have identical morphology with the multipods Fe₃O₄@SiO₂@RF&PMOs nanoparticles, indicating that SiO₂ middle layer of the Fe₃O₄@SiO₂@RF does not affect the surface topological structure of the final nanocomposites after multi-site nucleation and growth of PMOs.

2. When the PVP is used to modify the surface of Fe₃O₄@SiO₂@RF nanoparticles, a special Janus structure is obtained. The multi-site nucleation is suppressed by adding PVP. How does the PVP change the surface kinetics of the material? Whether the mechanism of PVP, amine group and nitro group regulating surface property is the same? Please discuss more about this.

Response: We appreciate the reviewer's suggestions. The mechanism of regulating the surface kinetics of RF by surface modification of polyvinyl pyrrolidone (PVP), amine group and nitro group, is almost the same. The surface kinetics of RF mainly depends on nucleophilic ability of functional group on its surface, which further determines the number of nucleation sites on the surface. Similar to the amine group, the nucleophilic ability of carbonyl group in PVP is also weaker than phenolic hydroxyl groups in RF. After the surface modification with PVP, the initial phenolic hydroxyl groups in RF is covered by carbonyl groups with weaker nucleophilic ability. Therefore, the surface kinetics of RF is decreased by PVP. So, the multi-site nucleation of PMOs is suppressed and Janus nanocomposites are obtained. Actually, not only PVP, other kinds of polymers with weaker nucleophilic group, such as

poly(sodium-*p*-styrenesulfonate) (PSS) also possess such function to suppress multi-site nucleation of PMOs on RF surface.

We have accepted the suggestion of the reviewer. Accordingly, we have added the following illustration on Page 6 in the revised manuscript:

Other kinds of polymers, such as poly(sodium-*p*-styrenesulfonate) (PSS) also possess such function to suppress multi-site nucleation of PMOs on RF surface (**Supplementary Figure S13**).

We have also added the following illustration on Page 13 in the revised manuscript:

The surface kinetics of the heterogeneous nucleation of surfactants/silicate oligomers on RF can also be precisely manipulated by the surface modification with polymers (PVP, **Figure 2C**). The nucleophilic ability of carbonyl group in PVP is weaker than phenolic hydroxyl groups in RF. After the surface modification with PVP, the initial phenolic hydroxyl groups in RF is covered by carbonyl groups with weaker nucleophilic ability. Therefore, the surface kinetics of RF is decreased by PVP coverage. The critical nucleation concentration of oligomers on RF-PVP surface is increased. Thus, the starting time of the heterogeneous nucleation is delayed. Duration time of nucleation is shortened and the multi-site nucleation of PMOs is hindered, resulting in Janus architectures.

We have also added Supplementary Figure S13 in Page S13 of the revised Supporting Information:

Supplementary Figure S13. The TEM image of the Janus Fe₃O₄@SiO₂@RF-PSS&PMO nanocomposites obtained from the growth of PMO on the poly(sodium-*p*-styrenesulfonate) (PSS) modified Fe₃O₄@SiO₂@RF nanospheres. Similar to PVP (**Figure 2C**), the surface modification with PSS can also hinder the multi-site nucleation of PMOs on RF surface, resulting in Janus architectures.

3. The linear correlation between the nucleation site number and surface area is given in the paper. How to understand the surface area and how to calculate it in this paper? The author said the mesopore channels can provide additional “active” surface area for the multi-site nucleation. Dose the author consider the surface

area provided by the mesopores when making the linear curve between the nucleation site number and surface area?

Response: We appreciate the reviewer's suggestions. In our manuscript, the Fe₃O₄@SiO₂@RF nanoparticles are considered as an ideal sphere. The outer surface area of nanospheres can be calculated according to the formula $S = \pi d^2$ (d is the diameter of the nanoparticles). The near linear correlation is concluded only for non-porous Fe₃O₄@SiO₂@RF nanoparticles, not the mesoporous Fe₃O₄@SiO₂@mRF nanoparticles.

According to our results, the presence of the mesopore channels indeed has significant effect on increasing the nucleation number of PMOs on RF surface (**Figure 4e, f**). However, for the Fe₃O₄@mRF nanoparticles with mesoporous RF outer surface, it is very hard to define the contribution of outer surface area to the multi-site nucleation, as not all of the mesoporous walls (including the walls near the exit of the channels and the walls in the deep channel) can be used as "active" surface area for the multi-site nucleation.

Accordingly, we have added the following illustration on Page 8 in the revised manuscript:

The Fe₃O₄@SiO₂@RF nanoparticles are considered as an ideal sphere. The outer surface area of nanospheres can be calculated according to the formula $S = \pi d^2$ (d is the diameter of the nanoparticles).

4. The author reported that the generation speed of the oligomers in the solution is quicker at the relatively earlier starting time of the nucleation on the RF surface than that of SiO₂ surface. Are there any characterizations and evidence for that?

Response: We appreciate the reviewer's suggestions. The generation speed of the oligomers in the solution is correlated to the remaining amount of unhydrolyzed silanes. The amount of unhydrolyzed silanes decreases due to their consumption into generated oligomers, thus the generation speed of oligomers decreases as reaction proceed. For both RF and SiO₂, the generation speed of the oligomers in the solution is the same under the identical reaction conditions and time. So, the earlier the starting time of nucleation, the higher generation speed of the oligomers. Additional experiments were conducted in the revised manuscript. The results clearly indicate that the starting time of the nucleation of PMOs on RF (T_1) is much earlier than that on SiO₂ (T_2). So, we can inversely deduce that the generation speed of the oligomers at T_1 (start nucleation on RF) is higher than at T_2 (start nucleation on SiO₂).

We have accepted the suggestions of the reviewer. Accordingly, we added the following illustration on Page 12 in the revised manuscript:

In comparison, the generation speed of the oligomers is slower at the relatively later starting time of the nucleation on the SiO₂ surface, because the generation speed in the solution is decreased as the prolonging of hydrolysis of the silane precursors,

which can be confirmed by the experimental results (**Supplementary Figure S28, S29**).

We have also added Supplementary Figure S28 in Page S20 of the revised Supporting Information:

Supplementary Figure S28. TEM images with different magnifications of tetra-pods $\text{Fe}_3\text{O}_4@\text{SiO}_2@\text{RF}\&\text{PMOs}$ nanoparticles obtained at (A, E) 10 min, (B, F) 20 min, (C, G) 40 min, (D, H) 60 min of the reaction. The multi-site nucleation of PMO pods can be observed at 20 min, indicating that the nucleation of PMOs on RF starts earlier than 20 min.

We have also added Supplementary Figure S29 in Page S20 of the revised Supporting Information:

Supplementary Figure S29. TEM images with different magnifications of Janus $\text{Fe}_3\text{O}_4@\text{SiO}_2\&\text{PMO}$ nanoparticles obtained at (A, C) 30 min and (B, D) 60 min of the reaction. Comparing to the multipods $\text{Fe}_3\text{O}_4@\text{SiO}_2@\text{RF}\&\text{PMOs}$ nanoparticle (**Supplementary Figure S28**), nucleation of PMO on SiO_2 started much later.

5. Check the whole manuscript carefully. There are some mistakes, such as “.....(c) 160 nm; (c) The linear fitting of the number of PMO cubes.....” in the caption of Figure 4.

Response: We appreciate the reviewer’s suggestions. These mistakes have been corrected and highlighted in the revised manuscript. We have accepted the

suggestions. We have also checked out the entire manuscript and corrected some other grammar mistakes, which are all highlighted in the revised manuscript.

Reviewer #2:

This manuscript is interesting which demonstrates the controllable fabrication of multipods mesoporous Fe₃O₄@SiO₂@RF&PMOs nanocomposites with precisely controlled surface topological structures. The obtained unique nanostructures and proposed surface kinetics mediated multi-site nucleation strategy are intriguing. The findings in this manuscript are crucial for rational design of mesoporous nanoparticles with controllable topological structure and bio-applications with enhanced nano-bio interactions. Overall, the work has been carefully performed. I recommend acceptance of this manuscript after the following comments are addressed.

Response: We appreciate the reviewer for the positive comments.

Comments:

1. Although the authors demonstrated the general applicability of the surface kinetics mediated multisite nucleation strategy for forming multipod surface topologies on various kinds of nanomaterials, it seems that ordered surface topology is achieved only on nanospheres (in the text of the manuscript), whereas for spindles, cubes and nanotubes (Figure S14), the grown pods are randomly distributed. The authors should give some discussion on this.

Response: We appreciate the reviewer's suggestions. We have accepted the suggestions. The uniform distribution of periodic mesoporous organosilica (PMO) cubes on the spherical nanoparticles is mainly due to the isotropy nature of nanospheres. The multi-site nucleation of PMO cubes may take place in an evenly distribution manner on spherical resorcinol-formaldehyde (RF) resin surface, which induces the formation of most stable and highest symmetric structure for the multipods nanocomposites. As for anisotropic nanomaterials, such as spindles, cubes and nanotubes, the nucleation site may vary on different sites of the anisotropic surface, therefore hindering the uniform distribution of PMO cubes.

Accordingly, we have added the following illustration on Page 9 in the revised manuscript:

However, unlike the uniform multi-site nucleation of PMOs on isotropic Fe₃O₄@SiO₂@RF nanospheres, the distribution of the PMO cubes on the spindles, cubes and nanotubes are not as uniform as the tetra-pods Fe₃O₄@SiO₂@RF&PMOs nanoparticles, which can be attributed to the anisotropic morphology of the spindles, cubes and nanotubes.

2. For the surface of the initial RF nanospheres without the PMO cube nucleation and growth, is the RF surface exposed or covered by organosilica? More discussion

on this point should be provided in the revised manuscript.

Response: We appreciate the reviewer's suggestions. We have accepted the suggestions. Due to the strong nucleophilicity of the phenolic hydroxyl groups on RF surface, the silane oligomers can deposit on to the RF surface. So, the RF surface is also covered by a very thin layer of the organosilicas. The main difference between such layers and PMO cubes is that the thin layer of organosilica does not have ordered mesostructure.

Accordingly, we have added the following illustration on Page 5 in the revised manuscript:

It's worth mentioning that the surface of the initial RF nanospheres without the deposition of PMO cubes is also covered by a very thin layer of organosilicas. The main difference between such layer and PMO cubes is that the thin layer of organosilica does not have ordered mesostructure.

3. The linear correlation between nucleation site number and particle size is well established in this manuscript. From the TEM images, it seems that the size of the PMO cubes is also quite related with the number of the cubes, which should be clarified.

Response: We appreciate the reviewer's suggestions. Indeed, the size of the PMO cubes is related to the number of cubes on the surface of the nanoparticles. We have measured the size and number of the PMO cubes on the $\text{Fe}_3\text{O}_4@\text{SiO}_2@\text{RF}$ spheres with a different size of 300 nm (**Supplementary Figure S15**), 260 nm (**Figure 1c**), 230 nm (**Figure 4a**), 200 nm (**Figure 4b**), 160 nm (**Figure 4c**). The results show that ~ 8 cubes are grown on the large 300-nm nanospheres. Uniform tetra-pods topologies with 4 PMO-cubes can be formed on $\text{Fe}_3\text{O}_4@\text{SiO}_2@\text{RF}$ with a diameter of 260 nm. In contrast, tri-pods (3 cubes), bi-pods (2 cubes) and mono-pod (1 cube) structured nanocomposites are obtained when the diameter of $\text{Fe}_3\text{O}_4@\text{SiO}_2@\text{RF}$ nanoparticles decrease to 230, 200 and 160 nm, respectively. The average side length of the PMO cubes decrease from 300 to 100 nm as increasing of the number of PMO cubes on the $\text{Fe}_3\text{O}_4@\text{SiO}_2@\text{RF}$ surface from 1 to 8, because the total amount of silane precursors is constant, the more cubes grown, the smaller the size of each one.

Accordingly, we have added the following illustration on Page 8 in the revised manuscript:

Beside the number of the PMO cubes, the average side length of the cubes decreased from 300 to 100 nm as increasing of the number of PMO cubes on the $\text{Fe}_3\text{O}_4@\text{SiO}_2@\text{RF}$ surface from 1 to 8. When the total amount of the silane precursors is constant, the more cubes grown, the smaller the size of each one (**Supplementary Figure S16**).

We have also added Supplementary Figure S16 in Page S15 of the revised

Supporting Information:

Supplementary Figure S16. The correlation between the size and number of PMO cubes on $\text{Fe}_3\text{O}_4@\text{SiO}_2@\text{RF}$ with different diameters. The average side length of PMO cubes decrease from 300 to 100 nm as increasing of the number of the PMO cubes on the $\text{Fe}_3\text{O}_4@\text{SiO}_2@\text{RF}$ surface from 1 to 8. The bars represent mean \pm s.d. derived from $n = 20$ randomly selected nanocubes. Source data are provided as a Source Data file.

4. The authors claimed that the number of nucleation site is related to the speed of oligomer consumption and generation in the solution, which can be tunned by manipulating the surface property of the nanoparticles for the nucleation. Theoretically, increasing the added amount of silane precursor can also increase the generation speed of silane oligomer, therefore affecting the number of nucleation site. So, I suggest the authors to provide some experiments to further support the proposed mechanism.

Response: We appreciate the reviewer's suggestions. We have accepted the suggestions. Theoretically, increasing the amount of silane precursors can increase the generation speed of silicate oligomers. However, the high concentration of silane oligomers in the solution may induce the additional homogeneous self-nucleation of oligomers in the solution to form PMO nanoparticles, rather than the heterogeneous nucleation on the RF surface. The control experiments are provided in the revised supporting information to support the proposed mechanism.

Accordingly, we have added the following illustration on Page 13 in the revised manuscript:

In comparison, although the generation speed of silicate oligomers can be increased by increasing the amount of the silane precursors and the alkalinity of the solution, such increase is invalid for the heterogeneous multi-site nucleation. The high concentration of the silane oligomers in the solution induced by increasing the precursor and alkalinity of the solution, can only result in the homogeneous nucleation of the oligomers in the solution, rather than the heterogeneous nucleation on the RF surface (**Supplementary Figure S31**).

We have also added Supplementary Figure S31 in Page S22 of the revised Supporting Information:

Supplementary Figure S31. TEM images of (A) the Janus $\text{Fe}_3\text{O}_4@\text{SiO}_2\&\text{PMO}$ nanoparticles by increasing the concentration of the silane precursor (BTTE) from 3.4×10^{-3} to 7.0×10^{-3} M. (B) the Janus $\text{Fe}_3\text{O}_4@\text{SiO}_2\&\text{PMO}$ nanoparticles by increasing the concentration of ammonium hydroxide from 0.55 to 1.0 M. It can be seen that the nucleation site number cannot be changed by increasing the concentration of the silane precursors or ammonium hydroxide. The poly-disperse Janus $\text{Fe}_3\text{O}_4@\text{SiO}_2\&\text{PMO}$ nanoparticles and pure PMO nanoparticles are observed in the samples.

5. Some other minor points:

(1) The concept of “multivalent interaction” is also used in design of drug molecule/antibody. The difference between this “multivalent” and the multivalent of nanoparticles should be provided in this article.

Response: We appreciate the reviewer’s suggestions. The concept “multivalent” in the design of antibodies refers to that antibodies have more than one fragments with binding domains, therefore being able to bind to several antigen determinants at the same time, increasing the strength of binding between the antibodies and its target. This “multivalent” means “multiple interactions through chemical bonds”. On the other hand, the “multivalent” in this article refers to the multipods nanoparticles interacting with biological hosts through the multiple pods. This is basically a Van der Waals' interaction, and affected mainly by the morphology of the nanoparticles.

We have accepted the suggestions of the reviewer. Accordingly, we have added the following illustration on Page 10 in the revised manuscript:

It should be noted that, the concept “multivalent” here is different from the “multivalent” that usually refers to increasing the strength of antibodies’ binding towards targets with more than one binding domains according to chemical bonding. The “multivalent” in this article refers to the multipods nanoparticles interacting with biological hosts through the multiple pods, which is mainly based on a Van der Waals' interaction, and affected by the morphology of the nanoparticles.

(2) In Figure 6b, the surface kinetics is attributed to the difference in nucleophilicity between Si-OH groups and oxygen anions of RF. I suggest using wider arrow to represent stronger nucleophilicity. The present illustration may be misleading, suggesting that the main difference is the OH group number.

Response: We appreciate the reviewer's suggestions. We have accepted the suggestions. The Figure 6b has been re-illustrated for better understanding.

(3) In Figure 5, the significant difference value needs to be provided.

Response: We appreciate the reviewer's suggestion. We have accepted it. The significant difference values are provided in Figure 5. Therefore, new Figure 5 is provided in the revised manuscript.

Reviewer #3:

This manuscript focused on the use of surface-kinetics mediated multi-site nucleation strategy to synthesize mesoporous multipods with different topological structures. The authors successfully demonstrated their ability to tune the surface topology of nanoparticles. I do not have any major concerns on the novelty with respect to the fabrication. Relatively speaking, a weakness of this paper is on the demonstration of potential applications (i.e., the section on bacterial adhesion and inhibition).

Response:

Thank the reviewer for the positive comments. We have accepted them and provided responses accordingly.

Comments:

1. Methodological details in collecting the TEM/SEM images – The authors are encouraged to 1) demonstrate representativeness of the images presented and 2) further explain the experimental details on how the collection of these images was conducted. For examples: How many images were taken per sample? How much variation was observed between images? In other words, it's important to demonstrate that effort has been taken to minimize subjective bias in imaging the nanoparticles.

Response: We appreciate the reviewer's suggestions. We have accepted them. The methodological details about collecting the TEM/SEM images are provided in the revised manuscript. TEM images were acquired by directly dropping a drop of bacteria suspension on the copper grid and then drying the sample for observation. The dried samples were sputter-coated with gold for SEM imaging. More than 10 positions are randomly selected to take the electron microscope images. The images of the same sample do not show significant variations. Actually, one of the most

important functions of the electron microscope is to provide an intuitional visual understanding of the interactions between the nanoparticles and the bacteria at the microcosmic level. Compared to the whole sample, the observed areas by electron microscope are very limited. So, the electron microscope cannot be used as the precise quantitative tool to determine the amount of nanoparticles adherent on the bacteria. In order to provide quantitative analyses of the nanoparticles adhered amount on bacteria, inductively coupled plasma (ICP) based on Fe^{3+} concentration is applied.

Accordingly, we have added the following illustration on Page 10 in the revised manuscript:

Quantitative analysis results by inductively coupled plasma (ICP) show that the adhesion amount of the multipods T-RF&PMOs is almost fourfold comparing to the control groups (**Figure 5c**), further evidencing the significance of the multipods topological structure enhanced nano-bio interaction.

We have also added the following illustration on Page 15 in the revised manuscript:

A drop of bacteria suspension was dropped and dried on the copper grid for TEM characterization. The dried samples were sputter-coated with gold for SEM imaging. More than 10 positions were randomly selected to take the electron microscope images. The quantitative analyses of the nanoparticles adhered on bacteria were determined by inductively coupled plasma (ICP) based on Fe^{3+} concentration.

We have also provided a few typical SEM images of the tetra-pods $\text{Fe}_3\text{O}_4@\text{SiO}_2@\text{RF}&\text{PMOs}$ nanoparticles adhered on *E. coli* surfaces on Page S18 in the revised Supporting Information:

Supplementary Figure S22. The typical SEM images of the tetra-pods $\text{Fe}_3\text{O}_4@\text{SiO}_2@\text{RF}&\text{PMOs}$ nanoparticles adhered on *E. coli* surfaces. The images do not show any significant variations.

2. Interpretation of TEM/SEM images – The authors are encouraged to exercise caution when interpreting these images. For examples: 1) on Page 12, the authors claimed that the nanoparticles “have weak interactions with the bacteria” and 2) on Page 14, “the anchoring strength of T-RF&PMOs nanoparticles on the bacteria is very strong and stable”. TEM/SEM does not provide information about binding

affinity. Using words like “strong”, “weak”, and “strength” is an over-interpretation of the image.

Response: We appreciate the reviewer’s suggestions. It can be concluded from the SEM/TEM images and ICP results that, the amount of tetra-pods structured nanoparticles adhered onto bacteria is higher than that of nanoparticles with the Janus and core@shell morphologies. We agree with the reviewer that the strength of the interaction between the nanoparticle and bacteria cannot be directly evaluated by the electron microscope images. We have accepted the suggestions of the reviewer. We have revised the related section to avoid the over-interpreting of the TEM and SEM images. The words like “strong”, “weak”, and “strength” are deleted.

Accordingly, we re-phrased the following illustration on Page 10 in the revised manuscript:

In contrast, J-RF&PMO and RF@PMO nanoparticles without the multipods structure are barely observed in the SEM and TEM images (**Supplementary Figure S23, S24**), indicating that they cannot be efficiently adhered onto *E. coli*’s surfaces. Quantitative analysis results by inductively coupled plasma (ICP) show that the adhesion amount of the multipods T-RF&PMOs is almost fourfold comparing to the control groups (**Figure 5c**), demonstrating the significance of the multipods topological structure enhanced nano-bio interaction.

We have also re-phrased the following illustration on Page 10 in the revised manuscript:

Since a large amount of nanoparticles are adhered on *E. coli*’s surface, the bacteria are easily separated along by the magnet under a magnetic field.

We have also re-phrased the following illustration on Page 11 in the revised manuscript:

Herein, the T-RF&PMOs nanoparticles with both high-surface-area mesopores for drug loading and unique tribulus-like tetra-pods structure for enhanced bacteria adhesion, are employed to antibacterial assessments (**Supplementary Figure S26**).

We have also re-phrased the following illustration on Page 11 in the revised manuscript:

The lysozyme-immobilized T-RF&PMOs nanoparticles can retain over 90 % bacterial inhibition efficiency with negligible *E. coli* growth even in a long term of 3 days (**Figure 5g**).

3. E. Coli incubation – It seems that the incubation of E. coli with the nanoparticles was done at high ionic strength (based on the content of LB medium provided in S.I.). Did these nanoparticles aggregate in such condition? If so, were the aggregation dynamics (extent and kinetics) different for the various types of nanoparticles? And how would aggregation affect their reactivity with E. Coli? If

there was no aggregation, what enables the stability of the nanoparticles?

Response: We appreciate the reviewer's suggestions. The surfaces of the nanoparticles are rich of hydroxyl functional groups, enabling their stability in aqueous solution. periodic mesoporous organosilica (PMOs) are a kind of materials which have been widely used for biological applications. Their stability has been demonstrated by numerous works (*e.g. Nat. Commun.*, **2019**, *10*, 1241; *Adv. Mater.*, **2018**, 1707612; *Adv. Mater.*, **2016**, *28*, 3235). In the revised manuscript, the stabilities of the nanoparticles in Luria-Bertani (LB), phosphate buffer saline (PBS), acid and basic solutions, *etc.* are evaluated by dynamic light scattering (DLS). The results show that the average diameters of the nanoparticles keep nearly unchanged in different solutions over a long period of times, indicating that no aggregation of the nanoparticles was occurred in these solutions. Digital photos of the LB solution with tetra-pods structured $\text{Fe}_3\text{O}_4@\text{SiO}_2@\text{RF}\&\text{PMOs}$ nanoparticles are also provided, the solution retains clear with none observable suspended matter over 12 h, indicating the good stability of the multipods nanoparticles in LB solution.

We have accepted the suggestions of the reviewer. Accordingly, we have added the following illustration on Page 5 in the revised manuscript:

The multipods nanoparticles are well dispersed in Luria-Bertani (LB) solution over 12 h, demonstrating a good stability (**Supplementary Figure S7**). The stability of the multipods nanoparticles in LB, phosphate buffer saline (PBS), acid and basic solutions, *etc.* are also evaluated by dynamic light scattering (DLS). The results show that the average diameters of the nanoparticles remain nearly constant in different solutions over a long period of times, indicating no aggregation of the nanoparticles in the solutions (**Supplementary Figure S8**).

We have also added Supplementary Figure S7 in Page S10 of the revised Supporting Information:

Supplementary Figure S7. Digital photos of Luria-Bertani (LB) solutions of the tetra-pods structured $\text{Fe}_3\text{O}_4@\text{SiO}_2@\text{RF}\&\text{PMOs}$ nanoparticles after different periods of times: (A) 0 h, (B) 6 h and (C) 12 h. The solutions maintain clear with none observable suspended matter, indicating the good stability of the multipods nanoparticles in LB solution.

We have also added Supplementary Figure S8 in Page S10 of the revised Supporting Information:

Supplementary Figure S8. Average particle diameter acquired from dynamic light scattering (DLS) measurements of the tetra-pods structured $\text{Fe}_3\text{O}_4@\text{SiO}_2@\text{RF}&\text{PMOs}$ nanoparticles, Janus structured $\text{Fe}_3\text{O}_4@\text{SiO}_2@\text{RF-PVP}&\text{PMO}$ nanoparticles and core@shell structured $\text{Fe}_3\text{O}_4@\text{SiO}_2@\text{RF}@PMO$ nanoparticles in a series of different solutions: (A) Luria-Bertani (LB), (B) Roswell Park Memorial Institute 1640 medium with 10 % fetal bovine serum (FBS), (C) pH = 4 and (D) pH = 8 solutions for different periods of times. These results show that the diameters of the nanoparticles remain nearly constant in different solutions, indicating no aggregations of the nanoparticles in these solutions. The bars represent mean \pm s.d. derived from $n = 3$ groups of nanoparticle suspension. Source data are provided as a Source Data file.

REVIEWERS' COMMENTS:

Reviewer #1 (Remarks to the Author):

I find that the authors have made sufficient efforts in this round to clear my concerns. I'm pleased to recommend its publication as is.

Reviewer #2 (Remarks to the Author):

I have read the authors' responses and their revised manuscript. In general my comments have been addressed in this revision. The quality of this work has been improved and the conclusions are supported by datasets. I recommend acceptance as it is.

Comments with regards to addressing reviewer #3's points:

In general the authors have addressed each comments, and added new experimental data discussions and experiments. The manuscript has been revised carefully. However, some minor technical details should be further provided according to Reviewer 3's original comments.

- 1) Please explain whether the particle concentration for dispersity test is the same as the bacterial incubation condition.
- 2) Please provide TEM and DLS result of lysozyme loaded particles to confirm whether the surface topological structure is maintained and responsible for enhanced bacterial adhesion after lysozyme loading.
- 3) One additional comment on a spelling mistake: Carbanyl group should be revised to carbonyl group.

Manuscript Title: Surface-Kinetics Mediated Unique Mesoporous
Multipods for Enhanced Bacterial Adhesion and Inhibition

Point-by-Point Response to Referees

Reviewer #1:

I find that the authors have made sufficient efforts in this round to clear my concerns. I'm pleased to recommend its publication as is.

Response: We thank the reviewer very much for the approval of the work.

Reviewer #2:

I have read the authors' responses and their revised manuscript. In general my comments have been addressed in this revision. The quality of this work has been improved and the conclusions are supported by datasets. I recommend acceptance as it is.

Response: We thank the reviewer very much for the approval of the work.

Comments with regards to addressing reviewer #3's points:

In general the authors have addressed each comments, and added new experimental data discussions and experiments. The manuscript has been revised carefully. However, some minor technical details should be further provided according to Reviewer 3's original comments.

Response: We appreciate the reviewer for the time and expertise provided.

1. Please explain whether the particle concentration for dispersity test is the same as the bacterial incubation condition.

Response: We appreciate the reviewer's suggestions. The particle concentration for dispersity test is the same as the bacterial incubation condition. We have clarified this in the revised manuscript.

Accordingly, we have added the following illustration on Page S4 in the revised Supporting Information:

Particle stability examination:

Three kinds of nanoparticles are dispersed in different solutions: Luria-Bertani (LB) media, phosphate buffer saline (PBS), acid and basic solutions with a concentration of 0.5 mg mL^{-1} . The particle concentration was the same as the bacterial incubation condition. The dispersion was shaken at 200 rpm and taken at certain time for dynamic light scattering (DLS) measurements.

2. Please provide TEM and DLS result of lysozyme loaded particles to confirm whether the surface topological structure is maintained and responsible for enhanced bacterial adhesion after lysozyme loading.

Response: We appreciate the reviewer's suggestions. TEM images and DLS measurement were taken for lysozyme loaded tetra-pods $\text{Fe}_3\text{O}_4@\text{SiO}_2@\text{RF}&\text{PMOs}$ nanoparticles. The results confirm that the surface topological structure and good dispersity in Luria-Bertani (LB) media is maintained.

Accordingly, we have added the following illustration on Page 11 in the revised Manuscript:

The surface topological structure and good dispersity of T-RF&PMOs is maintained after lysozyme loading (Supplementary Fig. 27).

Accordingly, we have added the following illustration on Page S18 in the revised Supporting Information:

Supplementary Figure 27. TEM images and DLS measurement of lysozyme loaded tetra-pods $\text{Fe}_3\text{O}_4@\text{SiO}_2@\text{RF}&\text{PMOs}$ nanoparticles. The results confirm that the surface topological structure and good dispersity in Luria-Bertani (LB) media are maintained after lysozyme loading.

3. One additional comment on a spelling mistake: Carbanyl group should be revised to carbonyl group.

Response: We appreciate the reviewer's suggestions. The spelling mistake has been revised.